# Sediment transport drives tidewater glacier periodicity

Douglas Brinkerhoff[1], Martin Truffer[1] & Andy Aschwanden [1]

Most of Earth's glaciers are retreating, but some tidewater glaciers are advancing despite increasing temperatures and contrary to their neighbors. This can be explained by the coupling of ice and sediment dynamics: a shoal forms at the glacier terminus, reducing ice discharge and causing advance towards an unstable configuration followed by abrupt retreat, in a process known as the tidewater glacier cycle. Here we use a numerical model calibrated with observations to show that interactions between ice flow, glacial erosion, and sediment transport drive these cycles, which occur independent of climate variations. Water availability controls cycle period and amplitude, and enhanced melt from future warming could trigger advance even in glaciers that are steady or retreating, complicating interpretations of glacier response to climate change. The resulting shifts in sediment and meltwater delivery from changes in glacier configuration may impact interpretations of marine sediments, fjord geochemistry, and marine ecosystems.

[1] Geophysical Institute, University of Alaska Fairbanks, Fairbanks, AK 99775, USA. Correspondence and requests for materials should be addressed to D.B. (email: dbrinkerhoff@alaska.edu)

Despite a globally consistent trend of glacier mass loss[1], ~1/3 of Alaska's tidewater glaciers are advancing[2]. This trend shows little spatial consistency, suggesting a dynamical rather than climate mechanism is responsible. Tidewater glaciers differ from their terrestrial terminating cousins because mass loss occurs not only by surface melt, but also by calving and ablation at the oceanic boundary. The mass budget of such glaciers is therefore sensitively dependent on conditions at the marine terminus such as water temperature, motion, and depth. One example of this dependence, first observed at Columbia Glacier[3], is that the presence of a terminal shoal can lead to a considerable terminus advance relative to a position in absence of sedimentation. Such glaciers also tend to be unstable: ice flux from calving increases with terminus depth, and on a retrograde slope (such as the upstream side of a moraine) this positive feedback can lead to catastrophic retreat[4, 5]. The coupling of nonlinear ice dynamics and sediment transport leads to a continuum of dynamical behavior known as the tidewater glacier cycle (TGC), which can be described in four archetypal phases[6].

In the advancing stage, development and advection of a shoal at the front reduces calving flux, causing glacier thickening and advance. The shoal may be subaqueous (e.g., Hubbard Glacier, advancing 35 m per year[7]) or subaerial (e.g., Taku Glacier, advancing 10 m per year[8], Fig. 1). Eventually, the glacier enters an extended phase, in which the balance of accumulation and ablation halts advance (e.g., Brady Glacier[9]). A glacier enters the retreating phase when the glacier can no longer maintain sufficient thickness to remain grounded on the shoal, and the associated reduction in basal drag leads to retreat into progressively deeper water, triggering the instability described above (e.g., Columbia Glacier, which began such a retreat in 1985 that continues presently[6], Fig. 1). Ungrounding results either from the ice thinning or from the bed lowering due to erosion or mass wasting. Retreat ends when the glacier approaches the terminus position it would assume in the absence of sedimentation, and the terminus effectively re-grounds on bedrock. This retracted state is usually many kilometers shorter than the advanced terminus position. A sediment shoal may then rebuild and the cycle begin again. The time scales of the TGC have been inferred from a combination of direct observation, radiocarbon dating, and tree-ring analysis at several glaciers in Alaska, and cycle periods range from a few hundred years[10] to a few thousand[11].

Zero-dimensional modeling efforts have shown that sediment deposited at the glacier front can initiate advance through the calving reduction mechanism defined above[12–14], even when highly simplified models of ice dynamics (e.g., volume-area scaling) and sediment transport are used. Such models are also capable of simulating the retreat phase, although the degree of hysteresis between the time scales of advance and retreat is not explicitly captured. Models were inconsistent about whether an external perturbation is required to initiate retreat. Models designed to simulate the TGC in temperate glaciers have invoked either glaciofluvial erosion[13] or sediment deformation[12] as a mechanism for transporting the terminal shoal. More physically complete models have been applied to the problem of coupled ice flow and sediment deformation, the primary mechanism of sediment transport for the relatively dry marine ice sheets[15]. Interestingly, such simulations were shown capable of producing unforced distinctly asymmetrical oscillations in ice extent with obvious analogies to the TGC, albeit over hundred thousand rather than 100 year time scales.

Here we test the hypothesis that TGCs can be explained solely due to the coupling of ice dynamics, glacial erosion, and fluvial sediment transport by running a series of numerical experiments with a new model that explicitly simulates both ice and sediment dynamics. Our approach builds upon previous work in two primary ways. First, we consider sediment transport due to glaciofluvial rather than deformational processes[15], and we include a physically consistent representation of both erosion and depositional processes using mass conservation[13]. We find that that this coupled model, when calibrated with observations, can produce all phases of the TGC without externally driven changes in climate and with a period and amplitude in general agreement with observations from the geologic record. These temporal and spatial scales are strongly influenced by the availability of meltwater, which drives the rate of sediment transport and subsequently controling the rate of glacier advance. This dependence is sufficiently strong that a change in meltwater availability due to climate warming may (perhaps counter-intuitively) trigger advance in tidewater glaciers that are currently stable or retreating. Although necessarily a simplification of the complete physical processes governing tidewater glacier dynamics, our results provide a basis for assessing how such natural variability in sedimentation and meltwater regimes could impact fjord ecosystems, interpretation of marine sedimentary records, and predictions of ice volume change.

## Results

**Calibration.** Our model's physical configuration is inspired by Taku Glacier, with a mean length of ~50 km, and a maximum elevation of 2200 m. Figure 2 illustrates the geometry, climate, and sediment transport rate 100 years after the grounding line reached its minimum following model initialization. Mean bedrock erosion rates were tuned to match the tectonic uplift rate, as well as observations from coastal Alaska[16]. Fluvial erosion rates were calibrated with repeat radar measurements from Taku Glacier[17]. Fluvial deposition rates were derived from theoretical settling velocities of sediment samples taken from boreholes at

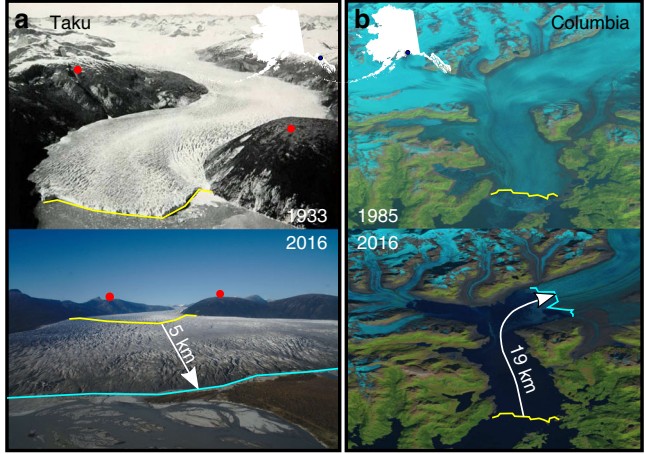

**Fig. 1** Advancing versus retreating glaciers in coastal Alaska. **a** Taku Glacier, which began advancing around 1850, is in the advance stage of the tidewater glacier cycle (TGC) (note the image obliquity). Image shows the terminus in 1933 (*yellow line*), when the glacier still possessed a vigorous calving front, versus 2016, where the glacier now terminates subaerially on glaciofluvial sediment (*blue line*). *Red dots* are the same location in each image. **b** Columbia Glacier, which began retreating in the mid 1980s, is in the dynamic retreat phase of the TGC. The *yellow line* is the terminus position in August 1985, near its maximum, whereas the *blue line* is the terminus position as of October 2016. Photo Credits: *top left*: U.S. Navy, 1929. Taku Glacier: From the Glacier Photograph Collection. Boulder, Colorado USA: National Snow and Ice Data Center. Digital media. *Bottom left*: M. Truffer. *Top right*: Landsat 7, data available from the U.S. Geological Survey. *Bottom right*: Landsat 8, Data available from the U.S. Geological Survey

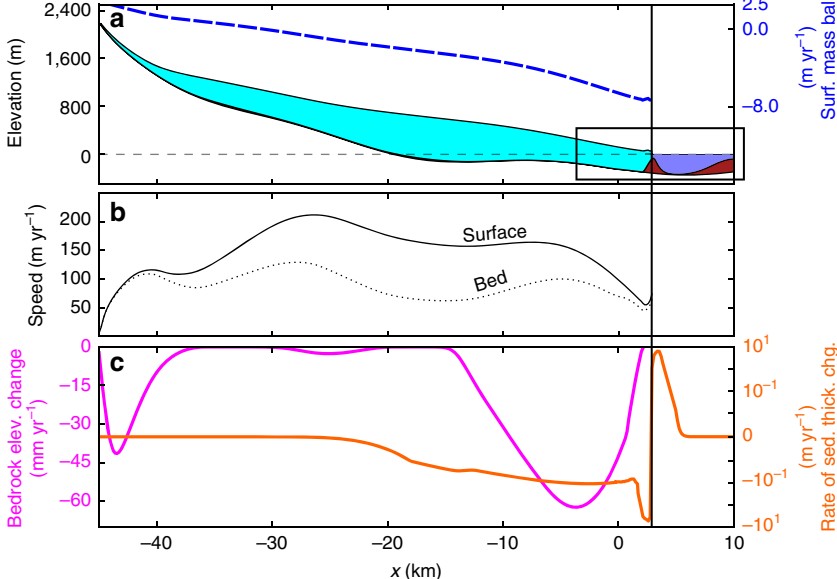

**Fig. 2** Model state during the advance phase of the TGC. **a** Geometric state of the glacier and surface mass balance rate (*blue hashed line*) typical of glaciers in coastal Alaska. *Black box* corresponds to extent shown in Fig. 3. **b** Surface and basal velocities. **c**. Sediment thickness rate of change due to fluvial transport (*orange*) and bedrock elevation rate of change due to basal sliding (*magenta line*), both of which were calibrated to roughly match observations from Taku Glacier in Southeast Alaska

Taku Glacier's terminus. The conclusions presented here are robust to parameter choices (Supplementary Note 1).

**The TGC in a temperate climate**. To establish our model's behavior under steady and contemporary climate assumptions, we run a temperate climate experiment with an elevation-dependent specific surface mass balance[18] approximating present-day coastal Alaska[19] where warm summer temperatures generate meltwater that causes high rates of glaciofluvial transport.

The evolution of the modeled glacier exhibits each phase of the tidewater glacier (Fig. 3; Supplementary Movie 1). The glacier begins in a stable and calving configuration ($t = 0$). Sediment is carried to the terminus by meltwater, forming a shoal. The shoal acts as a plug, reducing ice flux through the calving front and yielding a positive mass balance that prompts thickening and advance onto the shoal. The sediment composing the shoal is recycled and continuously transported downstream by glaciofluvial processes, growing larger as more is produced and delivered from upstream. The shoal eventually breaks the surface ($t = 240$ years) and calving ceases. The sediment that reaches the terminus at this stage is transported subaerially, depositing at the downstream end of a growing outwash plain. The glacier in this configuration is effectively land-terminating, albeit with a bed that continues to be reshaped by fluvial transport. As accumulation and ablation approach balance ($t = 286$ years), advance slows. However, advection of the shoal is not subject to the same length limitations imposed on ice by its tendency to melt. Sediment continues to migrate from the upstream side of the shoal past the terminus even after ice advance has ceased. It is this process, in which the rate of advance of the shoal outpaces the advance of the ice, that leaves the glacier unsustainably grounded in progressively deepening water, eventually initiating retreat. A void at the upstream end of the shoal opens ($t = 306$ years), rapidly increasing basal motion from 30 to 350 m year$^{-1}$ as the ice and bed decouple. Despite the longitudinal stresses imposed by the thin and rapidly deteriorating downstream, ice still grounded upon the shoal, more ice continues to reach the grounding line, triggering the irreversible positive feedback

between thinning and ice flux[4]. Catastrophic retreat ensues over 20 years, leaving the glacier in the retracted phase and ready to begin the cycle anew. We emphasize that the model produces this cycle of advance and retreat in the absence of any externally imposed climate change: the TGC can be understood as a limit cycle rather than a sequence of alternations between two unstable steady states driven by an external climate trigger, as has been previously hypothesized[13].

In the simulation presented here, the sediment supply was sufficient to produce a subaerial moraine, similar to those observed at Taku[17] or Brady[9] Glaciers. However, the shoal at Columbia Glacier when advanced did not breach the water surface. Reaching the endmember case of a subaerial moraine is not necessary for the mechanism proposed above: if the shoal is advected faster than the glacier can advance while still subaqueous, then the unstable retreat is still initiated. Indeed, in the case of increased fluvial transport (either through enhanced meltwater or more easily transported sediment) the period and amplitude of cycles is reduced for precisely this reason. In addition, more robust diffusion may keep the shoal from breaching the surface (Supplementary Fig. 1), potentially consistent with the relatively fluid sedimentary structures observed at Columbia Glacier[20].

Taking a more synoptic view, Fig. 4a shows the terminus position over 3 kyr. The glacier undergoes a sawtooth oscillation with an amplitude of 6.5 km and a period of 326 years, similar to the space and time scales of the TGC at Taku Glacier inferred from geologic evidence[10]. The cycle is superimposed upon a linear trend, which is caused by the progressive erosion of the submerged bedrock. As the bedrock is eroded landward, the stable position for the retracted phase migrates with it.

Total accumulation remains nearly constant through the simulation (Fig. 4b), as changes occurring near the terminus do not extend far enough to adjust the extent of the accumulation area. Periodicity is instead driven by the alternating importance of ablation and calving. Calving is moderate in the retracted phase, but decreases during the advance phase. Simultaneously, ablation is enhanced by increasing glacier area. The terminus reaches a maximum when accumulation and ablation balance while calving is zero, much like a

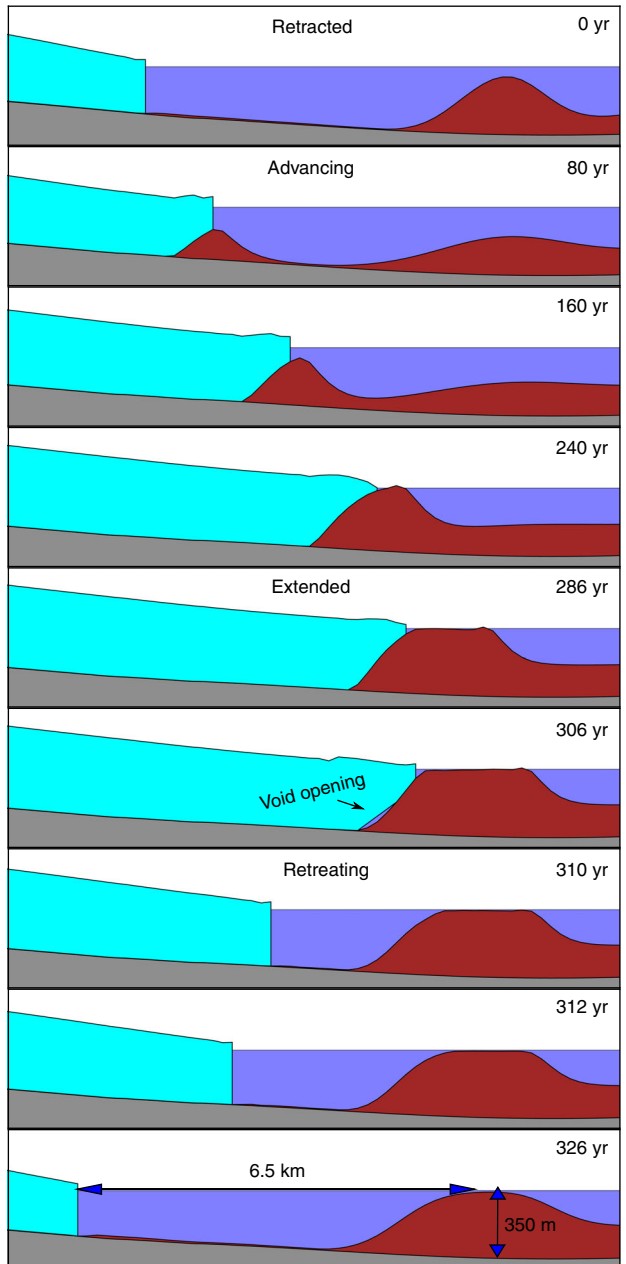

**Fig. 3** Modeled glacier geometry through a tidewater glacier cycle. Time period corresponds to the *shaded interval* in Fig. 4. *Cyan* represents ice, *blue* water, *brown* sediment, and *gray* bedrock. At each stage, the model solves the equations of ice and sediment motion described in "Methods". The *panels* are not uniformly distributed in time, due to the asymmetric time scales of advance and retreat. At $t = 0$ year the glacier is retracted, then advances until $t = 286$ years. At $t = 306$ years, fluvial erosion thins the shoal's upstream end causing the glacier to come afloat. The beginning of this process is evident in the small void developing at the upstream end of the shoal. Over the next 20 years, the glacier retreats towards its initial state

terrestrial glacier. However, calving rapidly increases to its maximum at the onset of the retreat phase, which combined with high melt leads to a rapid return to the retracted phase.

**The TGC in a cold climate**. At low latitudes, subglacial sediment transport is facilitated by copious melt[16]. Polar glaciers are comparatively dry, however[21], leading to the hypothesis that

tidewater glacier periodicity should be suppressed. We examine this by performing an experiment under climate conditions typical of arctic regions (low meltwater availability and relatively little snowfall). We find that the available meltwater cannot supply enough sediment to the terminus to overcome the shoal's background diffusion: hillslope processes move sediment away from the shoal faster than deposition can supply it over the time scales considered here. Consequently, the shoal stays mostly stationary, as does the terminus (Fig. 4a, $t < 766$ years). The meltwater regime under which fluvial sediment transport overcomes diffusion marks a bifurcation point at which sediment-driven advance begins (Supplementary Note 2; Supplementary Fig. 2). We also note that this scenario is transport-limited: although the colder climate produces lower mass turnover and reduced basal velocities relative to the temperate experiment, there remains a protective sediment mantle over much of the bed. In the case of very low basal erosion, periodicity would also be suppressed (Supplementary Fig. 1).

**The effects of enhanced surface melt**. We may also ask what the effect of enhanced surface melt from a warming climate will be. We first perform an experiment in which the melt rate increases over our temperate climate experiment (1 m year$^{-1}$ over 100 years) beginning mid-advance at $t = 766$ years (Supplementary Note 3; Supplementary Fig. 3), after which melt remains elevated. The cycle prematurely terminates, and retreat occurs half a century early followed by re-initiation of the TGC (Fig. 4a). Post-perturbation, the amplitude and period of cycles are both halved. Thus, glaciers advanced past the extended terminus position for the new melt regime are subject to tidewater instability. This is particularly relevant for glaciers such as Taku, which may enter the retreat phase earlier than would be expected under a stationary climate[17].

Applying the same perturbation to our cold climate experiment, fluvial transport, and glacier advance both remain negligible. However, a more marked effect occurs when applied to an intermediate climate (meltwater availability between cold and temperate climates). Prior to warming, the glacier advances very little (Fig. 4a). However, after the perturbation the glacier advances at the stately pace of 5 km over 2 kyr. This advance is unbounded for the geometry considered here, due to the positive feedback between elevation and surface mass balance. In reality, secondary constraints such as warm ocean water or a continental shelf break would act as a barrier to further growth. Imposing a larger perturbation of 2 m year$^{-1}$ also leads to a (faster) advance and the development of a TGC with a period of 1.4 ka and amplitude of 11 km. Thus glaciers in relatively warm polar regions such as Southeast Greenland or the northern Antarctic Peninsula may (counter-intuitively) advance in response to climate warming, albeit with a periodicity longer than in warmer climates. Such a scenario may also describe Svalbard's glaciers, but the resulting dynamics would be superimposed upon the polythermal surge mechanism exhibited there[22]. Analysis of the convolved signal is difficult because of an incomplete understanding of either contributing process.

**Discussion**
Our results show that a relatively simple ice-sediment feedback mechanism can explain internally generated cyclicity in temperate tidewater glaciers over centuries to millennia, with ramifications for the coastal landscape and ecosystem. Changes in location and configuration of the proglacial moraine alters the manner in which sediment is delivered to the sea. In high melt regions like coastal Alaska and Greenland, sedimentation can suppress benthic productivity and biodiversity, due to drowning

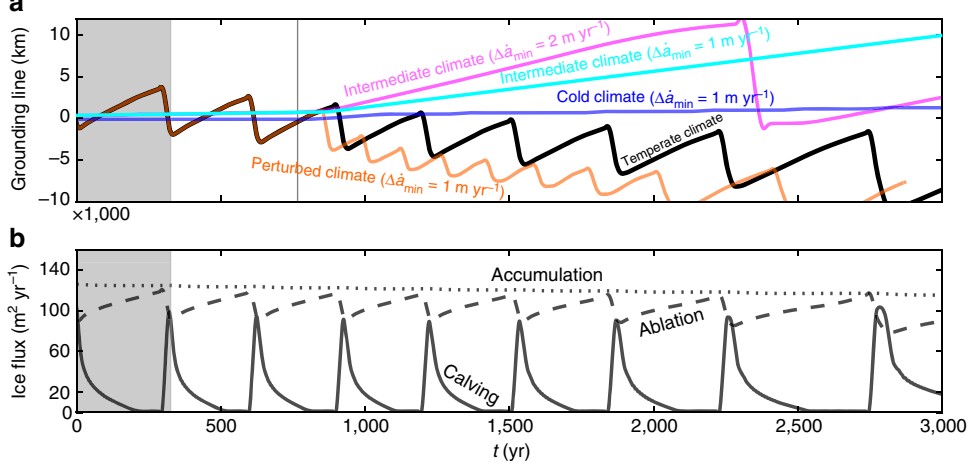

**Fig. 4** Grounding line positions and ice flux under different climate scenarios. **a** Position of the grounding line for the first 3 kyr of a 10 kyr simulation of the TGC. The *black line* shows the unperturbed temperate climate experiment. The *orange line* that partially overlaps, it shows the simulation in which the melt rate is increased by 1 m year$^{-1}$ beginning at the time indicated by the *vertical line*. The *dark blue line* shows this perturbed experiment but for a polar climate. The *magenta* and *cyan lines* show the perturbed experiment for an intermediate climate. Cycles continue past time frame shown. The decreasing trend in grounding line position, and the change in cycle length, is due to bedrock erosion and a change in bed shape. **b** Width-normalized fluxes due to net surface accumulation, net ablation, and calving for the temperate experiment

by marine snow and suppressed photosynthesis from turbidity[23]. However, sediments may also provide critical micro-nutrients further from the glacier front[24]. TGCs destabilize the sediment delivery mechanism: during the extended phase, sediment is delivered subaerially. Conversely, during the retreated phase, sediment is delivered near the sea bed.

TGCs also introduce periodicity in freshwater flux partitioning and fjord circulation. In the canonical advanced phase, all glacier mass enters the ocean as liquid runoff because the terminus is grounded near or above sea level and freshwater enters the ocean via subaerial streams. The introduction of freshwater at the surface promotes stable stratification. We note that this is still the case for ice fronts that still calve but are grounded in shallow water as was the case for Columbia Glacier before its retreat. On the contrary, during the retreated phase freshwater is injected at depth leading to vigorous buoyancy plumes, which promote fjord circulation[25]. Such circulation brings deep benthic and planktonic organisms to the surface where they become available to macrofauna[26]. Furthermore, invigorated circulation can also accelerate melting at the ice–ocean interface[27].

When retreated, tidewater glaciers produce icebergs, which act as critical habitat for pinnipeds[28]. Icebergs are advected by wind, and their freshwater mixes more slowly and over a broader area than does runoff, adjusting fluxes of glacier-sourced nutrients into and out of fjords[29, 30]. The chronic instability of sediment and freshwater fluxes may provide an important ecosystem service as waxing and waning of habitat suitability drives migration.

The reduction in calving flux induced by sediment-driven shallowing of the terminus allows for advance even in a warming climate, which may explain the continued advance of Hubbard, Taku, Yahtse, and other glaciers, despite the use of these phenomena as evidence repudiating climate change[31]. This mechanism may complicate regional studies of ice loss, which are important for predicting sea level rise and freshwater exports. However, shifts in climate have the potential to modify these cycles: an increase in melt enhances fluvial erosion and leads to faster, lower amplitude cycles, perhaps unto the point that they cannot be resolved. To the contrary, in cold regions, which lack the meltwater to produce TGCs, a warming climate may yield advance.

## Methods

**Model overview and empirical support**. Our numerical model solves coupled equations representing ice and sediment transport. To compute ice flow, we use an approximation to the Stokes' equations that accounts for both shear and membrane stresses, as well as a basal sliding that varies with changes in water pressure. Conceptually, ice is deposited by snowfall (which is a function of surface elevation), transported downstream by ice flow, and leaves the system either by calving (which essentially occurs when ice comes afloat) or by melt. Meltwater generated in this way is routed to the glacier bed and becomes available to move sediment, which is generated by the erosion from sliding at the glacier sole. We account for sediment transport due to fluvial entrainment and deposition, as well as gravitational diffusion. Sedimentation provides a feedback to glacial dynamics by adjusting the bed geometry, which alters the glacier's stress configuration and potentially causes ice to either ground or float. In an effort to maintain simplicity, we neglect many interesting but conceptually non-essential physical processes including but not limited to tectonic uplift, isostasy, oceanic heat transfer, and sediment deformation.

Repeat radioechosounding at Taku Glacier has shown net erosion rates as high as 4 m year$^{-1}$, far outpacing what could be expected based on sediment deformation[17]. This result is also supported theoretically[32] and by large scale observations indicating that temperate glacial environments are the most fluvially erosive on earth[16]. Indeed, transport of sediment through pure deformation is not likely to account for the shoal advance rates observed in meltwater-dominated tidewater glaciers[17, 33]. Depositional features directly observed in extant morainal shoals also suggest vigorous glaciofluvial transport[34].

**Ice flow model**. We use an ice flow model that simultaneously solves the equations of momentum and mass conservation (adapted from ref. [35]). We simplify the momentum conservation equations by using the Blatter–Pattyn approximation[36, 37], which assumes hydrostatic pressure and negligible vertical resistive stresses. Such a model is suitable for simulating both creeping flow and sliding. Separation of the ice from the bed occurs when hydrostatic ice pressure drops below water pressure. We do not use a specific calving law, instead imposing a strong basal melt term to floating ice, such that half its thickness is lost annually. In addition, we neglect lateral drag for floating ice. This crudely simulates the presence of near terminus effects such as ice mélange when the front is pressed against an obstacle such as a terminal moraine, and has no effect on upstream dynamics otherwise[5]. With respect to existing formulations, this represents an adaptation of the "calving on flotation" parameterization, with differences occurring only when the glacier comes afloat but is still pressed against a downstream obstacle. This approach is empirically supported by the fact that the model produces the correct time scales of both advance and retreat when compared with the observed tidewater glaciers. Furthermore, the qualitative behavior of the model is insensitive to how quickly melting occurs, with the model still exhibiting periodicity in the case where no melt occurs and a floating tongue is allowed to develop. We neglect changes in water density due to freshwater flux.

We use a generalized sliding law[38] with a correction for large bed slopes[39],

$$\tau_{b,i} = -\beta^2 (P_H - P_w)^{1/n} |\mathbf{u}_b|^{(1/n-1)} \mathbf{u}_{b,i} (1 - \mathbf{N}_i^2),$$  (1)

where $\tau_b$ is the basal shear stress, $\mathbf{u}_b$ the basal velocity, $\beta^2 = 6 \times 10^3$ Pa$^{\frac{n-1}{n}}$ m$^{-\frac{1}{n}}$ year$^{\frac{1}{n}}$

is a parameter chosen to yield a basal velocity, which accounts for roughly 50% of the surface velocity, $P_H$ is the ice overburden pressure, and $n = 3$ the Glen's flow law exponent (additional physical constants specified in Supplementary Table 1). **N** is the unit normal vector at the ice base. We assume a basal water pressure of $P_w = \max(0.7P_H, P_o)$, where $P_o$ is the water pressure imposed by the height of sea level. This effective pressure parameterization assumes good connectivity wherever ice is grounded below sea level. We do not differentiate between sliding over bedrock, sliding over till, or deformation of the till itself; rather we assume that the law described above parameterizes each of these types of basal motion.

**Sediment transport model.** We define the equations for the production and transport of sediment and water over the entire domain, including both glacierized and unglacierized areas. Sediment (with thickness $h_s$) is produced and bedrock elevation $B$ modified by glacial erosion mechanisms such as, quarrying, plucking, and abrasion, but we do not differentiate these. Instead, we assume that glacial erosion scales with the work done at the bed, which for our sliding law choice implies that it is non-linearly proportional to sliding velocity[40]:

$$\frac{dB}{dt} = -b \, \mathbf{u}_b \cdot \tau_b (1 - \delta_s). \tag{2}$$

Bedrock erosion efficiency $b$ is a function of bedrock properties and topography at unresolved length scales and cannot be directly measured. We tuned $b = 10^{-8} \, \text{Pa}^{-1}$ such that the spatially averaged bedrock erosion rate during our simulations is three times the tectonic uplift rate in southeast Alaska of 5 mm year$^{-1}$, and in strong agreement with observations[16]. The factor of three is utilized because we expect a higher bedrock erosion rate near the glacier centerline than in unglaciated or marginal areas (the influence of this and other parameters is explored in Supplementary Note 1). $\delta_s$ is an indicator of fractional sediment coverage that varies between zero at $h_s = 0$ m and unity at $h_s > 1$ m; when the bed is fully covered by sediment, bedrock erosion cannot occur. We neglect fluvial bedrock erosion because it is thought slow compared with the glacial erosion in tidewater glacier systems[16], though this assumption has been questioned[41].

We assume that sediment is fluvially transported. Water is provided by surface melt (which we assume immediately reaches the bed) and (to a much lesser degree) basal and englacial melt produced by sliding and deformation. These quantities are integrated along the hydraulic gradient to determine the total water flux $Q_w$

$$\nabla \cdot \mathbf{F}Q_w = [m_b - \min(\dot{a}, 0)], \tag{3}$$

where, **F** is the direction of water flow (here taken to be in the direction of the surface elevation gradient), $\dot{a}$ is the annual specific surface mass balance function, and $m_b = \frac{\tau_b \cdot \mathbf{u}_b}{\rho L}$ the basal melt rate, where $\rho$ is the density of ice and $L$ the latent heat of fusion. We note that seasonality is not included in our computation of meltwater: only net annual ablation is included in runoff calculations, which would tend to underestimate the overall amount of water available. We convert the flux $Q_w$ to a vertically averaged water speed $\bar{u}$ by assuming a characteristic subglacial drainage depth where grounded (henceforth 0.1 m) and water depth where floating to derive an effective water thickness $h_{\text{eff}}$. Conservation of mass then implies that

$$\bar{u} = \frac{Q_w}{h_{\text{eff}}}. \tag{4}$$

Net change in sediment thickness is governed by the mass conservation relation

$$\frac{\partial h_s}{\partial t} + \frac{\rho_r}{\rho_s} \frac{\partial B}{\partial t} = \dot{d} - \dot{e} + \nabla \cdot k \nabla (h_s + B), \tag{5}$$

where $\rho_r$ and $\rho_s$ are bedrock and sediment densities, and $k$ the diffusivity of sediment due to hill-slope processes, $\dot{d}$ the deposition rate, and $\dot{e}$ the fluvial erosion rate. We make the common assumption that the rate of fluvial entrainment of particles is proportional to stream power[42], which leads to the expression

$$\dot{e} = c \frac{\bar{u}^2}{h_{\text{eff}}} \delta_s, \tag{6}$$

where $c$ is the fluvial erosion efficiency, which we chose such that net erosion 200 m upstream from the terminus is ~2 m year$^{-1}$ and ~5 m year$^{-1}$ at 1 km, based on repeated radar measurements[17]. The deposition rate is proportional to the fluvial sediment flux $Q_s$ normalized by the water flux

$$\dot{d} = \hat{w} \frac{Q_s}{Q_w}, \tag{7}$$

where $\hat{w}$ is the fallout speed adjust by the distribution of sediment in the water column. We use $\hat{w} = 500$ m year$^{-1}$, the settling velocity of fine silt[43], which approximates the median grain size at the base of boreholes drilled by the authors near the terminus of Taku Glacier. We close the model with a second mass conservation equation accounting for mobile sediment, with the additional assumption that this quantity changes quickly relative to glacier time scales and the

time derivative can be neglected, yielding

$$\nabla \cdot \mathbf{F}Q_s = \dot{e} - \dot{d}. \tag{8}$$

**Parameterization of lateral variations.** Specifying the flow direction **F** through the subglacial hydrologic system is a problem that is operationally unsolved. Approximating it requires explicit models of subglacial hydrology, which we did not implement because they are both computationally expensive and possessed of many unconstrained and unobservable parameters. We instead sidestep this issue by making the simple assumption that water flows in the direction of the ice surface gradient, with a parameterized flow depth providing what amount to balance velocities. If we also assume a geometry that is axially symmetric and transversely uniform (with respect to the downstream direction) then we may easily width integrate the model, reducing computational time by collapsing the model's spatial extent to a single flowline. Such an assumption is certainly not valid over short length and time scales, with real glaciers exhibiting a great deal of flow-transverse heterogeneity in flux and efficient channels potentially transporting most of the water (and perhaps sediment) over a relatively small spatial footprint. However, we argue that our simulations are sufficiently long and the subglacial drainage configuration sufficiently variable, that averages of water and sediment flux over the glacier width capture the features salient to the purposes of capturing the interactions of ice and sediment at a large scale. A more theoretical justification of this assumption is outside the scope of this paper; instead, we appeal to heuristics by noting that the front of Taku Glacier exhibits a glacial outwash plane that extends across its entire terminus, despite extant major channels only exiting the glacier front in a few discrete locations.

In any case, this width-averaging has the effect of replacing the two-dimensional divergence operator $\nabla \cdot \mathbf{F}Q$ with the one-dimensional operator $\frac{\partial Q}{\partial x} + \frac{Q}{W} \frac{\partial W}{\partial x}$, where $W(x)$ is the glacier width. The additional term $\frac{Q}{W} \frac{\partial W}{\partial x}$ effectively accounts for transverse flow of mass into the cross section. We use the gamma distribution as a width function

$$W(x) = \frac{W_{\text{max}}^2 \chi^{-\kappa}}{\Gamma(\kappa)} (x + L)^{\kappa - 1} \exp\left(-\frac{x + L}{\chi}\right) + W_{\text{min}}, \tag{9}$$

where $W_{\text{max}} = 14$ km and $W_{\text{min}} = 3$ km are the maximum and minimum domain widths, $\theta = 5$ km is a length parameter, and $\kappa = 1$ is a shape function. We perform this lateral integration for the mass conservation equations of both ice and sediment. We do not account for lateral drag in the momentum conservation equations.

**Numerical methods.** We numerically solve the first order approximation of the momentum conservation equations[37] in width-integrated form using a linear Galerkin finite element method in the horizontal dimension. We split the domain into $n_e = 1000$ elements, for an average element size of ~$\Delta x = 75$ m. We discretize the vertical dimension using an ansatz spectral element method, in which we assume that the vertical velocity profile is adequately approximated by the linear combination of a zero-order and fourth-order polynomial[35].

We solve the depth-averaged form of the mass conservation equation to evolve the ice geometry through time. As such equations are nominally hyperbolic, we use a Streamline Upwind Petrov–Galerkin (SUPG) finite element method to stabilize the transport equation[44]. We discretize the mass conservation equation in time using the Crank–Nicholson method, which provides second-order accuracy and good numerical stability. We used a time step of $\Delta t = 30$ days for all simulations. This timestep guaranteed convergence at each time step for all experiments. Further refinement did not substantially change the numerical solutions. All finite element discretizations were performed with the FEniCS library[45].

The non-linear mass and momentum conservation equations were solved simultaneously using the SNES VI Newton solver from the PETSc numerical solver library[46]. This variational inequality solver allows for the specification of a lower bound on ice thickness to prevent the non-physical scenario of a negative ice thickness. In practice, we specified a lower ice thickness bound of 1 m to prevent numerical singularity in the momentum equations. Newton's method requires a Jacobian matrix, which we computed analytically using the automatic symbolic differentiation capabilities of the FEniCS library.

We simultaneously solve the equations of mass conservation for deposited sediment, transported sediment, and meltwater transport. The former is discretized using Galerkin finite elements and the latter two using the SUPG finite element method. Time discretization is accomplished with the Crank-Nicholson method, and numerical solution of these non-linear equations is performed with the PETSc SNES VI solver, once again utilizing the FEniCS library's capacity for symbolic differentiation. We specify a lower bound of zero on sediment thickness.

**Geometry and climate forcing.** The initial bedrock geometry varies between $B_{\text{max}} = 2200$ m at a sharp ridgeline, to sea level 30 km downstream. Here, the topography forms a fjord, which attains a maximum elevation of $B_{\text{min}} = 300$ m another 45 km out to sea. Sinusoidal bumps with a wavelength of $L/2$ were superimposed, where $L = 45$ km is a characteristic length scale. The topography

was given by the function

$$(B_{max} - B_{min})\exp\left(-\frac{x+L}{pL}\right) + B_{min} - A_s \sin\frac{4\pi x}{L}, \qquad (10)$$

where $p = 0.3$ is a shape factor and $A_s = 100$ m is the amplitude of topographic undulations.

In all experiments, climate is imposed through an elevation-dependent specific mass balance given by the function

$$\dot{a}_{min} + \frac{\dot{a}_{max} - \dot{a}_{min}}{1 - \exp(-f)}\left[1 - \exp\left(-f\frac{S}{B_{max} - B_{min}}\right)\right], \qquad (11)$$

where $f = 2$ is a shape function. The temperate climate experiment prescribes a maximum accumulation rate of 2.5 m year$^{-1}$ ice equivalent at 2200 m, and $-8$ m year$^{-1}$ at sea level, representative of conditions in Southeast Alaska (e.g., Taku Glacier[19]). The equilibrium line altitude is ~1000 m, and we impose no external climate variability. The cold experiment uses a $\dot{a}_{max} = 1.25$ m year$^{-1}$ and $\dot{a}_{min} = -1$ m year$^{-1}$. The intermediate climate has $\dot{a}_{max} = 1.75$ m year$^{-1}$ and $\dot{a}_{min} = -4.5$ m year$^{-1}$, respectively.

We begin each model run from a steady state in the absence of sedimentation (i.e. Eqs. 5–7 neglected and $h_s$ uniformly zero). Our criterion for steadiness is that ice volume changes by $<10^{-2}$% at each time step. After this initial state is computed, we allow the model to run through a single cycle of advance and retreat, and take the resulting sediment configuration as the initial condition for the model runs analyzed in the main body of the paper. We do this to eliminate transient signals associated with abruptly turning sedimentation on. In cases that do not exhibit periodicity over 10 kyr of model time, we initialize experiments from the 10 kyr state.

**Data availability**. The Python script used to perform these simulations is distributed with this manuscript as Supplementary Software. The authors encourage readers to study and utilize this script for verification or further experimentation. For computed model results, contact D.B. at dbrinkerhoff@alaska.edu.

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

## Acknowledgements

D.B. was supported by the NSF Graduate Research Fellowship US National Science Foundation (NSF) Graduate Research Fellowship Grant No. DGE1242789 and a Center for Global Change Student Research Grant. M.T. was supported NSF Grant Nos. PLR-1304899 and PLR-1443733. A.A. was supported by NASA Grant Nos. NNX13AM16G, NNX13AK27G, and NNX16AQ40G, and by NSF Grant No. PLR-1603799. We acknowledge Jason Amundson, Jesse Johnson, and Carl Tape, whose comments greatly improved this manuscript.

## Author contributions

D.B. performed all simulations, designed the study, and wrote this manuscript, all with substantial input from A.A. and M.T.

## Additional information

**Competing interests:** The authors declare no competing financial interests.

