## [Peer Review File · Nature Communications]

Reviewers' comments:

Reviewer #1 (Remarks to the Author):

Review of Brinkerhoff et al., *Sediment transport drives tidewater glacier periodicity*, submitted to *Nature Communications*

The major claims of the paper are namely that:

- (1) The advance and retreat of tidewater glaciers in regions such as Alaska and Patagonia display non-linear responses to climate variability, and are rather a function of sediment delivery and shoaling at the terminus that suppresses mass loss through calving.
- (2) The characteristics of these cycles (i.e. their periodicity and magnitude) are determined by the long term availability of meltwater, which in turn drives subglacial erosion/delivery of sediment to the terminus.
- (3) The timing of a climate perturbation (e.g. warming) within the advance phase of the tidewater glacier cycle (TGC) does not automatically result in retreat. This allows the counter-intuitive possibility of glacier advance in a warming climate, thus explaining the contemporary behaviour of Hubbard, Taku and Yahtse Glaciers (amongst others) in Alaska.

Summary

I find the results presented by the authors to be extremely interesting (within the framework they present), and an excellent example of how a relatively simple modelling approach can potentially explain fundamental behaviour. There are perhaps some issues with novelty (the modelling of an advance/retreat cycle is not novel, which the authors acknowledge), however the mechanism and the framework in which it is placed is. I remain to be entirely convinced of the mechanism, though I think the authors could do a better job of providing justification for it (see main points 1 and 2), while some results require some further verification (main points 2 and 3). The paper does have the potential to influence thinking in the field, though to increase the possibility of this the wider context and applicability of the findings to other tidewater settings need to be elaborated upon further (see main point 4). On the proviso of the authors addressing the points raised within this review (primarily the justification for the erosion/sedimentation mechanism presented), a resubmission should be encouraged.

The review will outline a few main points in the context of the paper's overall novelty and relevance. This will be followed by a few relatively minor points (mostly on presentation and structure).

Main points

- (1) Although the demonstration of cyclical behaviour with a self-consistent representation of erosion and deposition is novel, the overall result that sediment shoaling controls the slow advance and rapid retreat of a tidewater glacier is not (see Oerlemans and Nick, 2006; Nick et al., 2007). Given that the former is the main area of novelty that then provides the basis for the following experiments, I was surprised that the authors did not go into detail regarding the empirical evidence for this. Initially I was a little sceptical of the mechanism, though a bit of reading around the problem and the nature of the field sites helped persuade me that it could be feasible, but I remain to be entirely convinced.

I therefore recommend that the authors include a review of factors (either in the main text, SI or both) that would convince the reader of their erosion/sedimentation mechanism (e.g. the subglacial erosion found by Motyka et al. (2006), the reasons showing that glaciofluvial processes dominate subglacial erosion, evidence for rates of sediment supply to the terminus and the relationship to melt/subglacial runoff). I don't consider it enough to mention the evidence for the mechanism in fairly vague terms at different points in the text and refer the reader to different studies separately to let them reach their own conclusions (e.g. L50-1, L53-4 [this also needs a reference], L63, L80, L90) – the onus should be on the study to present a coherent justification for the processes that are built into the model.

Furthermore, on a more general theme, for those who are perhaps more familiar with shorter timescale observations, it would be beneficial to explicitly highlight examples of palaeo evidence for TGCs (both in Alaska and elsewhere) and the timescales they occur over.

Together, this would help go a long way to persuading similarly sceptical readers of the erosion/sedimentation processes included in the model without having to resort to a delve through the literature, and also provide crucial evidence and context that the mechanisms, rates of erosion and deposition, and the cycles themselves have empirical basis. If the authors are able to strengthen their case for their erosion/sed mechanism, then this will give the reader confidence that the results that follow are also valid.

The remainder of the review will operate on the assumption that erosion/sed is treated realistically within the model.

- (2) Leading on from the authors invoking subglacial hydrology as being key to building a shoal proglacially, some more background regarding what is known of subglacial hydrology beneath Alaskan tidewater glaciers would also provide useful context. There is a potentially non-negligible issue within this that the authors should address, in that the location of outlets for sediment laden subglacial water at the terminus will potentially be spatially limited (e.g. at the extreme case, if there is one portal for the entire ice front, focussing sediment delivery to one part of the terminus only).

Although there is some debate as to the distribution of subglacial discharge from tidewater glaciers at their termini, spatially limited sediment rich plumes are commonly observed, suggesting that the majority of meltwater (and therefore sediment deposition) is spatially concentrated as it enters a fjord. In this situation, the applicability of a width integrated model potentially falls down, given that shoaling will be asymmetric across the terminus.

To counter this, I would (ideally) like to see results of a short planform simulation showing whether advance behaviour can occur under such conditions of asymmetric shoaling. I appreciate that the authors mention in lines 201-203 that long planform simulations would be computationally expensive, but as long as one or two short simulations are not computationally *prohibitive*/the authors can comprehensively justify that such simulations are not needed, the results of these experiments would go a long way toward helping substantiate their mechanism for shoaling and glacier advance.

- (3) Given the highly seasonal nature of runoff, and that the advance mechanism is intrinsically linked to it through sediment supply to the terminus, there needs to be at least one set of sensitivity tests that captures annual runoff variability over a range of realistic values. This will demonstrate that the author's simplifying assumption that using annual runoff rates (that in turn contribute to sediment deposition at the terminus) is valid.
- (4) The discussion section as a whole needs to be strengthened substantially. Although I'm mainly a glaciologist, so perhaps I missed the importance of L118-134, they left me with the thought of "so what?", while I think L135-140 needs to include a wider relevance (i.e. implications for Greenland/Antarctica/other calving systems in general). There is some mention of this in L111-116, though I think this too needs expanding upon, most notably with respect to (1) the similarities and differences between Alaskan and other tidewater glaciers in terms of how analogous their climate and environments are; (2) to be more up front about how widely applicable TGCs may or may not be to different regions with tidewater glaciers; (3) what are the implications of these findings for global sea level

contributions/Alaskan Glaciers in the short term (i.e. <100 years) and the long term (>100 years and/or 1 TGC); and (4) highlighting (as mentioned in the abstract) that the public perception of Alaskan glaciers as bellweathers of climate might only be justified in the long term (i.e. >1 TGC/centennial timescales).

For example, with respect to points 1 and 2 in the previous paragraph, how applicable would the paper's findings be to areas where there is less uplift (and given the reasoning in lines L169-171, less erosion)? Given that this is arguably one of the major differences between Alaska/Patagonia and most other major regions with tidewater glaciers, I think this point is worth pursuing in order to ascertain the wider relevance of the findings. Although the answer to this is to some extent already in Figure S1b, the relevance of the TGC mechanism to this sort of scenario needs to be elaborated on.

Also, with respect to point 3, it's notable that in figure S2 the degree of advance following the perturbation appears to be the product of the integrated climate from the beginning of the TGC, which then impacts the amplitude of that particular cycle. Although it's an obvious point to make, it demonstrates that proglacial sedimentation does not insulate the glacier system entirely from climate. The authors already have sensitivity experiment results that would allow them to comment on the significance this to glaciers that are currently at different stages of their TGCs.

Minor points

- (1) The location of information relating to model set up, parameter choice, and forcing needs to be signposted much more clearly within the text. For example:
 - a. how calving is dealt with is not explicitly referred to in the text, but only mentioned in the Methods. Given that this mechanism is fundamental to controlling how the modelled glacier will advance and retreat, an explanation of the calving law implemented or a reference to where this can be found needs to be included in the main text. At present it has the feeling of it being hidden away in the fine print.
 - b. Same again, but for SMB, settling velocities of sediments, bedrock erosion rates, slope-driven diffusion, basal drag, the definition of polar, intermediate and temperate climates etc.All can be dealt with by just referring the reader to the methods/supplementary info in the right places

Also, in regard to model setup, every parameter mentioned that is held constant between runs should ideally be put into a table in the SI (physical constants as well as things like time step, grid size, ice temperature/flow law coefficient, flow law exponent etc). This will substantially help the reader get a better idea for how the model is set up at a glance.

- (2) None of the plots show glacier velocity, and neither is an impression given for the contemporary velocity structure of Taku Glacier/similar glaciers. It would be informative to show these (figure 3?) to highlight similarities/dissimilarities of the simulations with other tidewater glaciers globally.
- (3) L70 – how much does velocity increase by?
- (4) Figure 1 needs a location map showing where both Taku and Columbia are. If Taku is being used as the basis for the simulations (albeit in an idealised sense), the reader would benefit from what it looks like at a larger scale.
- (5) Supplementary discussion 3 – “an array of constants” – need to be more specific what is meant by this

- (6) Figure S3 would benefit from better labelling, and more detailed explanation in the caption in addition to a reminder of what Q_w/Q_0 is.
- (7) Supplementary methods: numerical methods – is there a reason why a timestep of 1 month was chosen? This seems fairly large for the simulation of potentially fast flowing, dynamic glaciers like Taku. Does refining the timestep change the overall results?
- (8) Supplementary code – it's excellent that the authors have included the code used to generate the results as it allows others to interrogate and explore the implications of their findings using precisely the same setup. The only recommendation I have for this is for the inclusion of comments mentioning which line of code refers to which equations in the methods section. This would be entirely for the extra convenience of the reader, rather than having to decipher the variable names in the code, or for situations where the reader is unfamiliar with python.

References

Motyka, R.J., Truffer, M., Kuriger, E.M. and Bucki, A.K., 2006. Rapid erosion of soft sediments by tidewater glacier advance: Taku Glacier, Alaska, USA. *Geophysical Research Letters*, 33(24).

Nick, F.M., van der Veen, C.J. and Oerlemans, J., 2007. Controls on advance of tidewater glaciers: Results from numerical modeling applied to Columbia Glacier. *Journal of Geophysical Research: Earth Surface*, 112(F3).

Oerlemans, J. and Nick, F.M., 2006. Modelling the advance–retreat cycle of a tidewater glacier with simple sediment dynamics. *Global and Planetary Change*, 50(3), pp.148-160.

Reviewer #2 (Remarks to the Author):

In this study, Brinkerhoff and others provide significant new and important insight into the physical understanding of the so-called "tidewater glacier cycle", which has been debated within the glaciological community for several decades. While this is admittedly not my specific area of glaciological expertise, I know enough about the subject to say that the results presented here are novel, intriguing, and will be of interest to a wide audience (e.g., glaciologists, climate scientists, geologists, geomorphologists).

Using a higher-order numerical model coupled to models of subglacial water flow, erosion, and sediment transport -- all calibrated to broadly reproduce the features of a prototypical tidewater glacier in S.E. Alaska -- the authors are able to show that coupled ice flow and sediment dynamics alone can reproduce all of the salient features of the tidewater glacier cycle. Of particular significance is that the coupled model is capable of doing so in the complete absence of any climate forcing, which to my knowledge is another unique finding of the study. A careful analysis and discussion of the model results provides the necessary insight for understanding the detailed physical mechanisms responsible for this behavior.

Overall, this paper is very well written and a pleasure to read. I would strongly recommend accepting it for publication.

Below, I have provided a few suggestions / comments / questions for the authors to consider. These are primarily aimed at improving the readability and clarity of the paper in a few places.

(numbers refer to line numbers in the original manuscript)

-- main body of manuscript --

4,5: Is the qualifier "in AK and Patagonia" needed?

8: "...reducing discharge into the ocean and causing GLACIER THICKENING AND advance ..."? (Is there an important implicit step missing here, i.e. glacier thickening?)

16: "...to trigger advance in glaciers". Make it explicit that these will then become tidewater glaciers?

18: Are there really examples of people using tidewater glaciers as "bellwethers for integrated climate"? My understanding was that only someone who knows nothing about glaciers would try to interpret their behavior in the context of monitoring changes in the broader climate.

28,29: Again, make it explicit that reducing calving flux leads to thickening which leads to advance.

32,33: "Such glaciers are subject to an unstable feedback ... retrograde slope." Would it be worth clarifying that this is, physically, related to the marine-ice sheet instability, even if only on a much more local scale?

In general, I think it could be stated a bit more explicitly, and early on, exactly what the mechanism triggering unstable retreat is (e.g., in lines 33-38, it would be nice to know what the "trigger" is, or how it operates to cause an unstable retreat). Is it sediment erosion on the up-glacier side of the shoal, which eventually separates the ice and bed? Is it some minor, localized

version of the instability that leads to the marine-ice sheet instability (i.e., retreat off the shoal into deeper water, increasing ice flux and thinning)?

41: Should this be listed as, e.g. "(methods)", rather than reference [8]? This is a bit misleading because the model used here is actually a variation on the model used in ref. 8. That variation is explained in the methods, but it is not really given by reference 8 on its own.

Is there the theoretical possibility of a configuration whereby the glacier and sediment dynamics conspire to give a steady-state terminus in the advanced position?

In Figure 2, you have the erosion given with negative units, which I think implies deposition. Shouldn't the erosion rate here be everywhere positive (since deposition is handled separately)? Either that, or the erosion rate could be expressed with negative units by giving it as the rate of bedrock elevation change.

62: Again, maybe explicitly mention the role that thickening plays in advancing the glacier.

62-65: Is subaerial erosion of the shoal accounted for?

In Figure 3, panel for 306 yrs, is the gap between the shoal and glacier here due to glacier thinning, erosion of the shoal, or both? This may be important to highlight in terms of the reader trying to get a handle on what the important physical process is that triggers the initial retreat from the shoal.

67-68: Sediment migrating from the upstream side of the wedge past the terminus sounds like a 3d effect. How does that happen in what is essentially a 2d model (since it get get around the sides of the wedge, or can it)? Is it being pushed up over the top of the shoal and then cascading down the other (ocean) side of the wedge?

69: Is the "lacuna" shown in Figure 3, e.g. the very small gap in the panel for yr 306? If so, it might be worth pointing it out explicitly?

81-82: Do you want to be even more explicit here and state that bedrock erosion will result in the stable position for the retreated phase of the glacier to retreat inland over time?

93-94: This could be clarified a bit. You mean essentially that the rate at which sediment is supplied to the shoal from the glacier (via meltwater) is not sufficient to grow the shoal (and hence the terminus cannot advance)?

103: This statement is a bit unclear. You mean that, based on historical records you expect that the Taku is going to start retreating around date X, but that in a warming climate, it will actually start to retreat at date $< X$?

126-129: Is the assumption here that in the advanced phase, all meltwater is entering the ocean at the ocean surface (i.e., because the glacier is grounded on a subaerial shoal)? If so, it might be good to state this explicitly. Intuitively that hadn't occurred to me, and seemed at odds with the statement of meltwater being injected at depth in the retreated phase.

137: Is there peer reviewed literature to reference here instead? It seems a bit disingenuous to reference this obviously biased website.

-- Methods ---

142: Make it explicit that you are using a modified version of the model presented in [8].

147: Is there some physical or observational basis for the choice of removing 1/2 the thickness annually?

155: Explain what the correction for large bed slopes does physically? Is it to account for things like cavitation (and hence ~ 0 friction) at the sides of steep bumps?

171: Which multiplicative factor do you mean here?

185-186: It's not clear to me why the overall melt would be underestimated by ignoring seasonality. Is there a non-linearity that should be pointed out?

When the gov. equations are explained in the methods, it might help to provide some discussion of the model domains. E.g., it wasn't clear to me where fluvial transport "ends". It cannot / does not occur in the fjord right? Is that defined by where the terminus of the glacier is?

189-190: Does Equation 5 hold everywhere at all times? It looks like $d_{\dot{}}$ is small or 0 under the glacier and that dB/dt is zero in front of the glacier. Trying to understand if this has to be defined over a time varying domain, or if it just operates everywhere at all times.

190: How well known is the "diffusivity" of the sediment due to slope processes? It seems like this is another parameter sensitivity that could be evaluated, since it will be critical for the evolution of the shoal (which in turn is critical for whether or not the glacier can continue to advance or is forced to start retreating).

192-193: In Equation 6, it seems like the expression in parentheses should be: $1 - (1 - \text{del}_s)$. I say this because if $\text{del}_s = 0$ for no sediment and $\text{del}_s = 1$ for $h_s \geq 1$ m, then as written Eq. 6 says there will be erosion when there is no sediment present and no erosion when there is sediment present (My understanding is that this equation refers to erosion of sediment not bedrock, since Eq. 2 covers the rate of bedrock erosion. That is, you want to erode bedrock when there is no sediment, and you want to erode sediment when the bedrock is covered by it rather than being exposed).

199: "mobile sediment"  "suspended sediment" ?

203: "...account for ICE flow convergence ..." ? Or are the equations for sediment, water transport, etc. also width averaged?

219: Perhaps it would be useful here to note what equations are turned "off" in the model to generate the initial condition?

222: "as our starting point"  "as our initial condition for the model runs analyzed in the main body of the paper"

-- Supp. Info --

parameter sensitivity:

As mentioned above, would it be useful to show or at least mention the model sensitivity to the diffusivity of sediment due to hill-slope processes? Or, is this a well constrained parameter? Since the important model behavior seems to be sensitive to the location / size of the shoal, it would seem that the model might be keenly sensitive to the choice of this parameter as well (e.g., for the same rate of sediment erosion and transport, a smaller / larger diffusivity might lead to a larger / smaller shoal, which would likely at a minimum affect the amplitude and period of the

advance / retreat cycles.

numerical methods:

- In paragraph 2, sentence 4, there is an "is" missing ("vertical profile IS adequately")
- 2nd to last paragraph, sentence 2 - "This variational inequality solver ALLOWS for ..."
- same paragraph, last sentence - "Newton's method requires a Jacobian matrix, WHICH WE COMPUTE ANALYTICALLY USING THE ..."

Reviewer #3 (Remarks to the Author):

This is a very interesting and in my view important study on the long-standing concept of the tidewater glacier cycle by Post that is here for the first time quantitatively assessing the full cycle and related dynamic feedbacks.

The concept of a cycle of rapid retreat through calving and slow readvance through sedimentation even in the absence of major changes in climate has been proposed on the base of observations several decades ago, but to my knowledge only once has partly been looked at with a quantitative model (Oerlemans and Nick..), that only looked at the aspect of readvance and lacked a dynamic model of the sediment deposition/erosion and related interactions with climate.

The paper here uses a state of the art flow model for calving glaciers that is coupled with a dynamic model for the evolution of basal topography including bed/sediment erosion, sediment transport which in turn are fully coupled to climate forcing (glacier melt water availability).

Applying this fully quantitative and dynamic model to an example close to a real world case of Taku glacier, this study is not only able to produce the hypothesized cycles in the absence of climate change but further demonstrates non-intuitive behaviour of such glaciers in relation to a changing climate (advance with warming climate) and interesting relationships between changes in period, amplitude and climate. The authors further provide a detailed sensitivity analysis to the model parameters and climate forcing that confirms their conclusions.

In my view, this is a very exiting and crucial study that demonstrates how such systems as tidewater glaciers behave very dynamically and that the interactions between several feedback mechanisms (involving calving, sediment erosion/transport/deposition, bed geometry evolution, surface melt) are producing very complex and highly non-linear behaviour in relation to climate.

Thus, this study provides a large step forward in better understanding these tidewater glacier systems and gives crucial new insights in interpreting current widespread observations of changing tidewater glaciers in a warming world. The high relevance of understanding tidewater glacier, but

also the integration of several sub-disciplines (glaciology, fluvial geomorphology/hydrology and landscape evolution, climate....) makes this publication accessible and interesting for a wide audience.

The quality of the manuscript/study is in general high and I identified only a few rather minor points/issues that I listed in more detail below. These points should in my view not question or weaken the general results and conclusions made. In brief they mostly concern some more clarifications/explanations of the methods and some clarification in explaining the modelling results and should be addressed before publication.

Further, I highly appreciate the provision of the model code, which lays fully open the approach used and is the ultimate step in reproducibility.

Overall, this is a high quality, highly relevant, clearly novel and original publication and I therefore very strongly support to publish this research for a wider audience in nature communication after addressing the minor comments/issues below. The list below appears pretty long, but this is just because I looked very much into the details, as I found it so interesting, and my criticisms on the paper are really very minor.

Andreas Vieli

More substantial points

1. Model explanations: Currently, when just reading the main text it is not always so easy to understand what the relevant feedbacks actually are and which process is governing the major changes (e.g. trigger for unstable retreat). When one reads the methods it gets clearer but they are pretty technical.

I guess this is partly an issue of the format of the journal which moves (for the sake of addressing a wide audience) the methods into a separate section outside the main text.

I think it would help to still explain in simple terms (few lines) the general elements of the model and their most important interactions (e.g. calving, erosion sediment transport/deposition, relation to climate/melt water) in the main text (at beginning of Model experiment/application/setup section) and in the very least a reference to the methods section should be given.

2. Methods:

There are few points that in my view need clarification:

-2a) calving model: I understand that here no explicit calving model is used (which I can accept and see the reason), but essentially by melting away the floating parts this translates into a flotation model (as for floating parts no lateral drag is applied, and under the assumption that hydrostatic pressure of the water is applied). Is this right? If so, this 'translation' would help as one can put it better into context of 'other existing' calving models. I assume the difference is when the floating part is still stuck at the shoal and experiences a longitudinal stress resistance. Related, what boundary condition is applied at the front (or below floating part?) hydrostatic pressure of the water?

-2b) width/lateral variations: I do not want to criticise the flowline approach, as this paper/modelling study is about the essential dynamic feedbacks/interactions along flow (so flowline fully appropriate).

But as all is presented as width averaged, it is not clear to me what width of the glacier/channel is actually used or whether it is constant along flow. Should be clarified as the area/width really plays a role if thinking of sediment production and available for deposition.

Further for the fluvial erosion of sediment I am wondering if the derived equations are still fully valid in a width averaged sense. I imagine subglacial channels to be very efficient to erode locally but not sure how to translate this into width averaged erosion. Maybe a few more explanations would help.

-2c) frontal boundary condition and water density change: what stress boundary condition is applied at the calving front? The standard vertically integrated balance between longitudinal stress and hydrostatic pressure balance which leads (using Glen's flow law) to a strain rate boundary condition?

I ask this because then the water density occurs as well and if I understand right the water density should change from ocean (advance stage) to freshwater when shoal reached surface and for initial phase of retreat (see fig 2). Is such a change in density considered in the model and relevant here?

-2d) isostatic adjustment: I assume there is no isostatic adjustment in the model but loads would change quite a bit and the bed adjust to it. It would probably only change the rates of advance and retreat (and perhaps reduce the general retreat trends and erosion could be compensated by uplift).

3. Sediment evolution and trigger for retreat:

sediment evolution is done purely through erosion (fluvial), transport within melt water and deposition, and so no sediment deformation is used here. This means retreat is triggered even in the complete absence of climatic change/fluctuations which is think is not the case if one would just use sediment deformation to advance the shoal because it would not be able to deepen the bed when approaching flotation at the front (and thus would require fluctuations around a mean to initiate enough retreat to get into unstable retreat mode). This difference or potential implications to using fluvial sediment erosion versus sediment deformation could perhaps be briefly discussed/mentioned.

4. Explanation of modelling results:

Maybe I am picky here but I struggled at times to fully follow the explanations of the modelling results and in particular explanatory feedbacks.

One specific point which I think I do not fully understand but which seems crucial for the whole paper is the specific reason/process of triggering the unstable retreat or in other words how the switch from advance to rapid retreat actually works.

I take from the paper that this happens when the total accumulation input is balanced by the surface ablation (as for land-terminating), so it can not advance any further but the fluvial erosion continues and erodes the upstream part of the shoal further until it starts floating there. Once it floats the calving model (melt beneath floating part) erodes away the ice and (due to deepening upstream) leads to unstable retreat. in the supplement (p. 5 second paragraph) this seems to get clearer when saying the advance of the shoal outpaces the front advance.

Now, for other parameter choices (bed/sediment erosivity suppl. Disc. 1, ...) the advance distance (cycle amplitude) is much shorter which likely means the glacier does not reach this balance in mass budget (balance flux zero at terminus) but still retreat is initiated (see supplements). And maybe the front does not get fully land-terminating. Why does it still initiate retreat? Is it simply that the erosion rate just behind the shoal is higher than the ice supply/thickening near the front and flotation is initiated? An/or is 'land-terminating' a necessary condition for initiating retreat. I think a better and more differentiated explanation of the exact mechanism and condition for retreat initiation would be useful. Maybe some labels/arrows... for the relevant subfigures in fig 3 may help.

5. Representativeness: Somewhat related top point above, the modelling experiments presented here are for a approx.. Taku Glacier which is fine but this is a case where the fjord gets fully filled up with sediment and hence the glacier land-terminating.

Some brief discussion of how representative the results are for other cases, like Columbia, which are not/never really land-terminating (e.g. no full fill up of fjord) would perhaps be useful, or at least I would be interested if the Columbia case can also be reproduced.

Specific (mostly minor) comments

p. 1 line 4: 'many' are advancing. i am not sure if there are that many, there are certainly some, but most are retreating or stable, or am I wrong? But 'some' or 'a number' may be more appropriate.

P1.lines 21 to 38: somewhere here the general effect of increased calving with water depth should be stated as it is essential to understand the concept, explanations and cycle here. on line 33 the effect of retrograde slope is mentioned but this requires exactly this relation of increased calving with water depth (to whatever power). Otherwise it is very hard (for non-experts) to understand why a shallowing water depth leads to advance or why the front retreats unstably. I would add a sentence for clarifying this near the beginning of the paragraph.

p. 1 line 38: '...advance restart...': should it not rather say '...restart advance...' or '..restart the cycle...'?

p. 3: line 46 and paragraph below: under the title 'Model' I would expect a model description but this is in the methods.

This is rather the 'Model experiment' or 'Model application'. Further, for better understanding how the model works I would briefly outline the crucial components (calving, sediment erosion/deposition, relation to melt water,...) and certainly refer to the methods section (e.g. for detailed model descriptions see Methods).

P. 3 line 57: basically the surface mass balance is elevation dependent, maybe indicate this here.

p. 3 line 63: 'pushed forward' seems not the appropriate term here, as there is no sediment deformation considered and the shoal is advancing simply by erosion and deposition. So 'moving forward' maybe more appropriate.

Same line: a word missing '...as more SEDIMENT continues...'?

p. 3/5 lines 67/68: 'Sediment migrates through fluvial transport from the upstream side...'

p. 3 line 72: '...between thinning, ice flux and eventually retreat.'

p. 6 lines 126-134: I assume this effect is not in model (applied melt rates) and a hypothetical implication (maybe clarify this). An again, is this change from ocean water to freshwater in the model/boundary condition at the front?

Methods:

p. 7 line 147/148: so essentially almost like a flotation criteria (see main comment 2a)

p. 8 line 187: discharge depth is uniform over whole width? so basically a sheet of 0.1m thickness? Seems a lot.

p. 8 line 193: '...which we CHOSE such that...'

p. 8. Line 199: I do not quite get why this mass conservation of sediment mass should be 'supplemental'. Seems a natural condition to me.

p. 8 lines 205: so how does the width vary in the used example here? or is it uniform? Seems important for sediment budget.

p. 8 line 207-211: I assume this bed topography/geometry is close/approximative to the case of TAKU glacier.

p. 8 line 212/213 eqn (10): seems pretty complicated mass balance function. Is this far from a linear function with elevation or what is the general form with elevation.

Fig 1: Are the images for Columbia at an angle (so not bird view)? I probably would mention it. Caption: there are no 'yellow dots, probably means 'red dots and yellow lines'. Further correct 'THE yellow line ...while THE blue line...'

Figure 2: I would add (a) and (b) in figure as for the bottom panel it is actually referred to 3b. In the bottom panel: are these the potential erosion or thickness change rates if sediment or bedrock is exposed/accessible. Further a vertical fine line may help at the position of the calving front in (b) in order to see where erosion/deposition is positioned relative to it.

Fig. 3: related to main comment 2b: strictly speaking during retreat it should be 'lake water' (fresh), is this considered in the model?

Figure 4: caption: 'THE black line...'; 'THE blue line...'; 'THE green line...'

Supplements:

Caption to video: maybe clarify that STANDARD temperate climate experiment (not obvious when just saying 'temperate' experiment).

Otherwise really helpful animation.

P. 4 table S1: what is the viscosity prefactor b ? in Fog. S1 later b is the bedrock erosivity.

p. 5 4th line from bottom: '...upstream from the sediment shoal,...can be deposited on the UPSTREAM side.' I do not get where this second UPSTREAM side should be.

P. 6 Suppl Disc. 2:

Line 1: what sign does the climate perturbation have (warmer/colder) or is this not relevant?

Line 6: 'conclusion' should be singular.

p. 7 Suppl. Disc. 3: line 3-5: '...by an array of constants...' I do not quite get how the climate is varied here, is just the mass balance gradient varied but the total surface mass balance (integrated along glacier) for the same geometry constant? What does this have to do with a tidewater environment?

Line 10: '...as quantified by VARIATIONS/SPREAD in grounding line position...'

p. 8 line 1: '...we NUMERICALLY solve these...'

Author Response for *Sediment transport drives tidewater glacier periodicity*

Douglas Brinkerhoff, Andy Aschwanden, Martin Truffer

1 Response to reviewer 1

1.1 Main Points

- 1. Justifying the mechanism** We have added a much expanded literature review covering the existing modelling work (including a paper by Alley c. 1991 salient to this subject of which we were previously unaware) in the Introduction, as well as the observational evidence supporting glaciofluvial transport as the dominant mechanism in these systems to a model overview section of the Methods. While we don't believe an in-depth discussion of geological evidence for TGCs is appropriate here, we have acknowledged it in the context of establishing length scales and added some references.
- 2. Subglacial hydrology** Constraints on subglacial hydrology, particularly in the map plane are not currently sufficient to justify the type of modelling that the reviewer suggests, in the sense that any new insights gained would be overshadowed by the uncertainty induced by the model itself. Thus we have decided not to include the proposed addition to the paper. However, we have included a paragraph in the results section attempting to justify why our simplified treatment may be adequate based on the variability in subglacial hydrology, the time scales involved in modelling these processes, and observations of proglacial geomorphology.
- 3. Seasonal runoff simulations** As with all modelling studies, the number of possible experiments that could be run is limited only by computational time. In this case, we fail to understand how doing sensitivity tests on our runoff parameterization would strengthen this paper, given that we have already shown simulations under a variety of assumed meltwater regimes.
- 4. Strengthening the discussion** We are also 'mainly glaciologists', but we believe the potential ecological ramifications are among the most interesting aspects of glaciers in AK and around the world. As a whole, the glaciological community has done a very poor job of making the case for

glaciers being relevant apart from a source for sea level rise. Frankly, these few glaciers that seem to undergo tidewater glacier periodicity are mostly irrelevant with respect to that issue, so we have ignored it. Instead, we have chosen to emphasize the relevance of tidewater glacier periodicity through discussion of the more local scale coupling between physical and biological processes, which we argue is not only scientifically important but also provides an additional mechanism by which to argue the importance of this work to a lay audience.

We do not fully understand the reviewer’s criticism regarding not emphasizing the broader likelihood of TGCs, when a significant portion of our results section is dedicated to applying experiments to cold climates and assessing the sensitivity of TGCs in these systems to climate change. Nonetheless, while our simulations are transport limited, even in the cold-climate case, we have included a few sentences about what would happen in the low erosion limit.

Regrettably, we do not understand the point suggested by the reviewer in the last paragraph here, particularly what the reviewer means by ‘the product of the integrated climate from the beginning of the TGC’.

1.2 Minor Points

- 1a. Since Nature Communications requires methods to be sequestered to the end, they tend to feel like so-called fine print, particularly for modelling papers. Nonetheless, our choice of calving parameterization is described there, and has also been elaborated upon.
- 1b. Writing ‘See methods’ throughout the manuscript seems an inelegant solution to the basic critique that the methods section appears in a non-optimal location for a modelling paper. Perhaps the reader should just assume that if they have a methodological question, then that material is likely available in the methods section.
2. **Glacier velocities** Since fluxes and hence velocities are essentially determined in a long term sense by the distribution of surface mass balance, then if our surface mass balance function is reasonable (which we believe it is), then the velocities must also be reasonable. We are not certain what utility reporting modelled velocities would have with respect to the narrative presented in the paper. We are not trying to recreate a specific glacier, rather we are trying to use modelling to support a more general hypothesis about an idealized mechanism.
3. **Velocity Increase** Added that basal velocity near the terminus increases from 30 m/yr to 350 m/yr.
4. **Location Map** We have included small location maps for both Taku and Columbia glaciers

- 5. **Suppl. Disc. 3** Reworded for clarity.
- 6. **Fig. S3** Added explanation of symbols.
- 7. **Suppl. Methods** Not a particularly large time step when using a time stepping scheme that is $\mathcal{O}(\Delta t^2)$ accurate. This is the largest time step we could use and still get textbook convergence at each time step for all simulations. Using smaller time steps does not appreciably affect the results. Added this explanation.
- 8. **Suppl. Code** Added equation numbers corresponding to text.

2 Response to reviewer 2

L4–5 Deleted qualifier.

L8 Added suggested phrase.

L16 We meant that steady glaciers that are already tidewater could begin to advance. Terrestrially terminating glaciers (in the classical sense) are not subject to the mechanism we discuss here.

L18 This is a fair point: credible people haven't been doing this for some time (although if one examines the internet literature of climate change skepticism, advancing glaciers being used as evidence against warming is fairly common). We've changed the line to reflect the general difficulty of assessing glacier response to climate.

L28–29 Changed 'growth' to 'thickening and advance'.

L32–33 I make this point when presenting this material as a talk, so it should probably be mentioned here. Added a discussion, and a reference to Schoof (2007). More generally, we've included a bit more about the possible retreat triggers.

L41 Agreed, changed to reference methods.

Is there a theoretical possibility of steady state terminus in advanced position?

We cannot provide a full answer to this question, but are inclined to say no under reasonable assumptions. Since the model isn't analytically tractable, It is difficult to try to find potentially unstable fixed points, so we cannot make a compelling mathematical justification. However, we can say heuristically that subglacial deposition is not favored by the model, and such a process would have to occur robustly to supply enough sediment to the terminus after it has reached its maximum.

Figure 2 Changed figure labelling.

L62 Reworded to emphasize this point.

- L62–65** Subaerial transport is accounted for (same model, but with a zero ice thickness and no meltwater input). Added a sentence that makes this clear.
- Figure 3** Most definitely due to the thinning of the shoal, which is one of the key points of the paper, and the reason that climate is not required to trigger a retreat. Emphasized this point in the figure caption.
- L67–68** It is being pushed over the top of the wedge. We have no mechanism for out of plane transport.
- L69** Emphasized this in the figure caption, and added a panel reference in the text.
- L81–82** Stated this more explicitly.
- L93–94** Clarified this point.
- L103** This is the correct interpretation. We have clarified this point in the text.
- L126–129** Again, the reviewer’s interpretation is correct. We added a more explicit statement of the processes we envision.
- L137** Our point for including this reference was to show how without a nuanced understanding of glacier physics, glacier advance yields an apparent contradiction to the simplified narrative of glacier retreat that can be used to attack science at large. Amongst the scientific community, the website is obviously biased, but it likely does not appear so the lay reader.
- L142** Made this clear. We hope that any confusion due to adaptation is ameliorated by the code being distributed with the paper.
- L147** No, not really, just a convenient mechanism for simultaneously simulating calving on flotation combined with the fact that ice doesn’t necessarily leave the system immediately when stuck behind sediment.
- L155** The correction for large bed slopes recovers first order accuracy for sliding, which is only really evident when the Blatter-Pattyn equations are derived from a variational principle. The Dukowicz reference explains this pretty well.
- L171** Clarified.
- L185–186** In actuality, the surface mass balance is given by $\dot{a} = \text{Snowfall} - \text{Melt}$, and what we should really be using to compute meltwater flux is $\max(\dot{a}, 0)$. However we’re just taking the negative part of \dot{a} , which assumes that Melt is zero whenever \dot{a} is positive, and that no precipitation falls below the equilibrium line altitude. This always underestimates runoff. However, the resulting errors end up being accommodated by the calibration procedure.

Domains All equations are defined over the entire domain at all times. It's just that water velocities are relatively high under the glacier so erosion dominates deposition, and very low in the ocean (because of the large depth) so deposition dominates. Added a note at L164 clarifying this.

L190: Diffusivity of the sediment is not at all well known. Worse yet, it may follow a multimodal distribution where fine sediment is highly diffusive, whereas coarse sediment is hardly diffusive at all. The suggestion of evaluating model sensitivity to diffusivity is a good one: we have included this new analysis in the supplement.

L192–193 This was a typo in Eq. 6. Corrected now.

L203 All equations are width averaged.

L219 Added.

L222 Changed.

Sensitivity Ran the suggested sensitivity tests.

Par. 2, Sent. 4 Corrected.

Par. -2, Sent. 2 Corrected.

Par. -2, Sent. -1 Corrected.

3 Response to reviewer 3

3.1 Substantial Points

Model Explanations We have added a paragraph under the Model heading which describes the essential functions of the model in conceptual terms and their feedbacks, with a reference to the more mathematical treatment in Methods.

Methods: Calving model We have added a sentence comparing what we've done here to the 'calving on flotation' model, and discussed how differences are only expected when the glacier comes afloat near a shoal.

Methods: Width We have added a paragraph in the methods section clarifying that we width integrate all of the equations, including those governing sediment. We also attempt to more clearly lay out the assumptions that go into width-integration, and also potential justifications for those assumptions.

Methods: Frontal boundary condition We don't really apply any boundary condition at the front *per se*, since we don't explicitly track the front, instead applying a water pressure condition to the ice base. The base does slope upward sharply downstream of the grounding line, approximating an

ice cliff. The change in density between freshwater and saltwater (a 2.9% change over a very small submerged distance) is certainly not sufficient to lead to meaningful differences in the solution. We have added a note clarifying this.

Methods: Isostatic adjustment This is certainly true. If we were more careful about trying to reproduce specific advance and retreat rates, we would have to require both isostasy and tectonic uplift. However, we felt that since our work is more intended to show an idealized result that including these features wouldn't have been worth the additional model complexity. Indeed, there are a variety of processes that we could have tried to include, simply because they did not seem essential for demonstrating the fundamental results of this paper. We have added some explanation of this in the Model heading.

Sediment evolution and trigger for retreat On the contrary, according to Pollard and DeConto (2009) who show similar unforced oscillations over 100ka timescales from simulations in Antarctica. We agree that the initiation of retreat would have to be more difficult, as sediment deformation could only ever bring the system to an unstable fixed point, while another process (though perhaps this could be numerical instability or diffusion) would have to push it over the edge. In any case, these processes must occur at fundamentally different time scales, and the results of Motyka (2006) make clear which one is more likely to be operating in a temperate environment. We had hoped to include a discussion of this paper in our literature review, but had cut it for space when submitting to Nature. We are glad to have the opportunity to include such a discussion.

Explanation of physical process We've tried to be a little clearer in the description of modelling processes in the results section. We've also added a more thorough caption and annotated Figure 3.

Representativeness With respect to trying to explicitly show a Columbiaesque scenario in which the land-terminating phase doesn't really start, that is indeed what happens when erosion is faster: the shoal outpaces the glacier before the glacier reaches a steady state (these processes are magnified by the thickness dependence of the surface mass balance). We have added a discussion of this in the results section.

3.2 Minor Points

L4 Changed to 'some' in the abstract, and gave a more precise description in the intro where citations are allowed.

L21–38 Restructured intro to introduce this topic more clearly

L38 Changed wording

L57 Mentioned that specific mass balance is elevation-dependent

- L63** We agree, changed to ‘continuously transported downstream by glaciofluvial processes’.
- L126** This mechanism doesn’t really have any physical implications without including submarine melt and oceanic circulation: the ocean is pretty big compared to the amount of runoff generated, and the density difference is small.
- L147/148** See response in **Methods: Calving model**.
- L187** Perhaps this is too large, but error in this parameter gets absorbed in the process of calibrating the erosion rate.
- L193** Corrected
- L199** We agree, and updated the wording.
- L205** Added a statement about width variability
- L207** It approximates Taku glacier in terms of characteristic scales of length and thickness, though likely not at all in terms of specifics.
- L212/213** Not too complicated: It’s linear when the shape parameter $\lim_{c \rightarrow 0} c$. Otherwise it just tends to flatten the mass balance gradient as we get to high elevations.
- Fig. 1** Corrected
- Fig. 2** Added a and b. They are actual rates. Added vertical line.
- Fig. 3** Eliminated word ‘ocean’
- Fig. 4** Corrected
- S table** Switched from b to A and clarified that this is associated with the momentum balance.
- P5, L-4** Clarified what we meant here.
- P6. Suppl. Disc.2 L1** Added that we mean a warmer perturbation.
- P6. Suppl. Disc.2 L6** Corrected.
- P7. Suppl. Disc.3 L3–5** Reworded to make more clear that we multiplied the entire mass balance field by a constant. In the absence of calving the length of the glacier should thus be the same. However, in a tidewater environment, the greater thickness of the temperate case means that it can extend further into the water before coming afloat.
- P7. Suppl. Disc.3 L10** Corrected
- P8. L1** Corrected.

REVIEWERS' COMMENTS:

Reviewer #1 (Remarks to the Author):

Left comments for the editor only.

Reviewer #2 (Remarks to the Author):

The authors have done a good job of addressing all of my previous comments. As I already thought the manuscript was very good, I don't have any other major comments / concerns at this time.

Minor Comments

- As the authors have labeled / identified the subglacial "void" due to erosion in the panel associated with year 306 of Figure 3, it might also be useful to explicitly point out and label the "lacuna" at the surface in that same figure panel.
- At the start of the methods, you have written "Our numerical model solves coupled equations representing ice and sediment transport (See Methods)." The parenthetical at the end is a bit strange since we are already in the methods section of the paper.

Reviewer #3 (Remarks to the Author):

Note that I reviewed this paper already the first time round and highlighted there in detail the novelty, importance, significance and scientific quality of this paper and which is even more valid now.

The authors have addressed the issues/questions I raised very carefully and I think this applies also to the points made by the other 2 reviewers as well. The methods are now really clear and the discussion and explanations of the mechanisms and model results much stronger and clear. thus overall this paper was now a real joy to read and I am pretty sure this paper will have a high impact regarding understanding and future research on the dynamics of calving glaciers in context of climate change.

I have only some very few points concerning figure 4:

- regarding the labelling in the figure of the orange line: for consistency (to other lines)I would label it as perturbed temperate climate ($\Delta a_{dot} = 1\text{m/y}$), so one can immediately see that this is a temperate but warming experiment.

- In the caption I would say (line 4) DARK BLUE LINE (instead of just BLUE) and more crucially I think the PERTURBED (also referring to blue line, line 4) should there be deleted, because this experiment (see reference to line 141) refers just to the cold climate (without any warming yet).

-nothing crucial, but rather to avoid misreading the figure: I would make a note in caption for Fig. 4a) that positive Gr-line values mean retreat and negative advance (as intuitively negative values are often taken as retreat).

Andreas Vieli

Author Response for *Sediment transport drives tidewater glacier periodicity-revised*

Douglas Brinkerhoff, Andy Aschwanden, Martin Truffer

May 8, 2017

1 Response to reviewer 1

Reviewer 1 has also requested that you include velocity data on your plots, please also make this change. We have included an additional panel showing both the surface and basal velocity in Fig. 2.

2 Response to reviewer 2

As the authors have labeled / identified the subglacial "void" due to erosion in the panel associated with year 306 of Figure 3, it might also be useful to explicitly point out and label the "lacuna" at the surface in that same figure panel. We used 'void' and 'lacuna' synonymously. We have changed all instances of the latter to the former to avoid confusion.

At the start of the methods, you have written "Our numerical model solves coupled equations representing ice and sediment transport (See Methods)." The parenthetical at the end is a bit strange since we are already in the methods section of the paper. Just the kind of embarrassing typo that frequent reformatting tends to produce. We have eliminated this reference.

3 Response to reviewer 3

regarding the labelling in the figure of the orange line: for consistency (to other lines)I would label it as perturbed temperate climate ($\Delta\dot{a} = 1$ m/yr), so one can immediately see that this is a temperate but warming experiment. We have added consistent notation to all of the lines in which the temperature perturbation was included.

In the caption I would say (line 4) DARK BLUE LINE (instead of just BLUE) and more crucially I think the PERTURBED (also

referring to blue line, line 4) should there be deleted, because this experiment (see reference to line 141) refers just to the cold climate (without any warming yet). Changed to dark blue line, but this line does refer to a perturbation experiment (the unperturbed experiment is just a flat line. The perturbed experiment is almost the same). We have changed the notation to emphasize this.

nothing crucial, but rather to avoid misreading the figure: I would make a note in caption for Fig. 4a) that positive Gr-line values mean retreat and negative advance (as intuitively negative values are often taken as retreat). In this case, positive grounding line values do mean advance, and negative means retreat. Perhaps the confusion stems from the counterintuitive phenomenon of advance in the intermediate perturbation experiments.